# Mechanism and Kinetics of Malachite Dissolution in an NH$_4$OH System

**Alvaro Aracena [1,*], Javiera Pino [1] and Oscar Jerez [2]**

[1] Escuela de Ingeniería Química, Pontificia Universidad Católica de Valparaíso, Avenida Brasil 2162, Valparaíso 2362854, Chile; javiera.p.rocco@gmail.com

[2] Instituto de Geología Económica Aplicada (GEA), Universidad de Concepción, Casilla 160-C, Concepción 4070386, Chile; ojerez@udec.cl

[*] Correspondence: alvaro.aracena@pucv.cl; Tel.: +56-322-372-614

**Abstract:** Copper oxide minerals composed of carbonates consume high quantities of leaching reagent. The present research proposes an alternative procedure for malachite leaching (Cu$_2$CO$_3$(OH)$_2$) through the use of only compound, ammonium hydroxide (NH$_4$OH). Preliminary studies were also carried out for the dissolution of malachite in an acid system. The variables evaluated were solution pH, stirring rate, temperature, NH$_4$OH concentration, particle size, solid/liquid ratio and different ammonium reagents. The experiments were carried out in a stirred batch system with controlled temperatures and stirring rates. For the acid dissolution system, sulfuric acid consumption reached excessive values (986 kg H$_2$SO$_4$/ton of malachite), invalidating the dissolution in these common systems. On the other hand, for the ammoniacal system, there was no acid consumption and the results show that copper recovery was very high, reaching values of 84.1% for a concentration of 0.2 mol/dm$^3$ of NH$_4$OH and an experiment time of 7200 s. The theoretical/thermodynamic calculations indicate that the solution pH was a significant factor in maintaining the copper soluble as Cu(NH$_3$)$_4$$^{2+}$. This was validated by the experimental results and solid analysis by X-ray diffraction (XRD), from which the reaction mechanisms were obtained. A heterogeneous kinetic model was obtained from the diffusion model in a porous layer for particles that begin the reaction as nonporous but which become porous during the reaction as the original solid splits and cracks to form a highly porous structure. The reaction order for the NH$_4$OH concentration was 3.2 and was inversely proportional to the square of the initial radius of the particle. The activation energy was calculated at 36.1 kJ/mol in the temperature range of 278 to 313 K.

**Keywords:** malachite; carbonate; leaching; ammonium hydroxide; heterogeneous model

## 1. Introduction

### 1.1. Consumption of Sulfuric Acid Due to Impurities

Copper oxide compounds are often treated using hydrometallurgy, specifically through the use of chemical leaching with acidic leaching (dissolution) solutions composed mainly of diluted sulfuric acid (H$_2$SO$_4$). However, when copper oxides contain a large quantity of carbonates (CaCO$_3$, MgCO$_3$) or hydroxides (Al(OH)$_3$, Ca(OH)$_2$), acid consumption increases enormously, to a level that makes metallurgical treatment economically inviable [1,2]. This consumption is mainly because carbonates as well as hydroxides are quicker to react with sulfuric acid than the copper oxides, because they are very soluble in acids [3,4], while in some cases the copper compounds can contain considerable amounts of carbonate and hydroxide in their crystalline system. This is the case with basic copper carbonates, such as azurite (Cu$_3$(CO$_3$)$_2$(OH)$_2$) and malachite (Cu$_2$CO$_3$(OH)$_2$). This leads to the complication

of compounds that excessively consume leaching reagent ($H_2SO_4$), making the leaching process inefficient and hindering copper recovery. There have been different works that have shown excessive consumption of acid with malachite; Bingöl [5] worked with a malachite mineral and obtained high copper recoveries (90%), along with other impurities, generating a consumption of 450 kg of acid per ton of ore. The same author tried to analyze the dissolution kinetics of malachite ore, but unfortunately he could not use a heterogeneous kinetic model (because the chemical reaction was very fast), obtaining a complete dissolution in a very short time caused by the present impurities that excessively consumed sulfuric acid (a group of minerals including pyroxene, quartz, goethite and magnetite, among others). Instead, Nicol [6] worked on the dissolution kinetics of malachite with $H_2SO_4$ (0.033 to 0.15 mol/dm$^3$), finding that the kinetics was governed by the chemical reaction on the surface. This could be achieved because he worked with malachite without impurities (i.e., acid consumers). In his work [6], he did not find the consumption of $H_2SO_4$ per ton of malachite.

An alternative option for the treatment of copper oxide minerals containing carbonates is to use leaching in an alkaline system, i.e., in an ammonium system. The main objective is to decrease acid consumption so that the process becomes more economically viable. In addition, the use of an ammoniacal system promotes dissolution selectivity as well as the reduction of corrosive attacks. Aracena [7,8] treated copper oxide minerals in an ammonium hydroxide system ($NH_4OH$). The experimental work was conducted using a stirred system with controlled temperature. The oxidized copper compounds such as tenorite (CuO) and cuprite ($Cu_2O$) were of high purity. The results obtained showed that copper can be extracted from tenorite (particle size of 5 μm; 0.45 mol/L (mol/dm$^3$) $NH_4OH$; pH = 10.5; temperature of 298 K; time of 300 min) and cuprite (particle size of 5 μm; 0.10 mol/L (mol/dm$^3$) $NH_4OH$; pH = 10.5; temperature of 318 K; time of 240 min), up to recovery values of 98% and 82%, respectively. The reaction mechanisms established in each study were the following:

$$2CuO + NH_4OH + 3NH_4^+ \rightarrow 2Cu^{2+} + 4NH_3 + 2H_2O + OH^- \tag{1}$$

$$2Cu_2O + 8NH_4OH + O_2 + 8NH_3 \rightarrow 4Cu(NH_3)_4^{2+} + 4H_2O + 8OH^- \tag{2}$$

The kinetic model representing tenorite and cuprite leaching was a chemical reaction on the surface. The activation energies calculated for tenorite and cuprite were 59.0 and 44.36 kJ/mol, respectively.

### 1.2. Leaching of $Cu_2CO_3(OH)_2$ Through the Use of Ammoniacal Systems

Several studies have been carried out with malachite leaching using an ammoniacal system. Ekmeyapar [9] studied malachite leaching using ammonium nitrate solutions ($NH_4NO_3$) and varying the working conditions of $NH_4NO_3$ concentration, particle size, stirring rate and temperature. The tests were carried out in batches, and the most significant results showed that, at a temperature close to 70 °C (343 K), copper recovery of 98% was obtained after 75 min (4500 s). The ammonium nitrate concentration was 4.0 mol/L (4.0 mol/dm$^3$). The reaction mechanism proposed by the researchers is given by the following expression:

$$Cu_2CO_3(OH)_2 + 4NH_4NO_3 \rightarrow Cu^{2+} + 4NH_3 + 4NO_3^- + CO_2 + 3H_2O \tag{3}$$

It was concluded that the reaction of the malachite with the ammonium nitrate followed a mixed-kinetics control, comprising two sequential mechanisms: for a temperature range of 30 to 50 °C (303 to 323 K), the leaching rate was controlled by the chemical reaction, giving $E_a$ = 95.10 kJ/mol; for temperatures of 50 to 70 °C (323 to 343 K), the dissolution rate was controlled by diffusion in the porous layer, as shown in the activation energy, which was 29.50 kJ/mol for this temperate range.

Bingöl [10] showed the effects of variables such as the ratio between the two reagents, pH, temperature, stirring rate, solid/liquid ratio (S/L), particle size and leaching time, on the leaching of malachite in ammonium hydroxide and ammonium carbonate solutions. The experiments were conducted in a batch system. The results showed that the optimal ratio for leaching malachite was

using a mixed solution of 5.0 M (5.0 mol/dm$^3$) NH$_4$OH and 0.3 M (0.3 mol/dm$^3$) (NH$_4$)$_2$CO$_3$, with a leaching time of 120 min (7200 s), temperature of 25 °C (298 K), stirring rate of 300 rpm and particle size below 450 μm, obtaining copper recovery of 98%. Finally, the researchers indicate that the dissolution kinetics of the malachite in ammonium hydroxide with ammonium carbonate was controlled by transfer on the interface and diffusion in the porous layer, obtaining an activation energy of 15 kJ/mol.

Künkül [11] performed experiments with a malachite mineral in a magnetically stirred reactor, analyzing variables such as particle size, ammonium concentration, solid/liquid ratio and temperature. Künkül found that by increasing the concentration of ammonium and the temperature while decreasing the solid/liquid ratio and particle size, a high-copper solution was obtained. The most effective parameter was the particle size (−125 μm). The porous layer diffusion model (produced by the SiO$_2$ that was around the malachite) represented the dissolution, with an activation energy of 22.338 kJ/mol being found.

Arzutug [12] used NH$_3$-saturated water to leach malachite mineral. The leaching experiments were carried out in a 250 mL (0.25 dm$^3$) glass reactor equipped with gas inlet and outlet tubes. The results showed that malachite reached its maximum dissolution (approximately 96%) for a particle size between +75–90 μm, temperature of 45 °C (318 K), ammonia concentration of 7.68 mol/dm$^3$, solid/liquid ratio of 2/100 g/mL (2/0.1 g/dm$^3$) and stirring speed of 400 rpm. Arzutug [12] found that malachite leaching was well represented by the second-order pseudokinetics with an activation energy value of 85.16 kJ/mol.

Studies related to the recovery of copper and iron from malachite minerals have been conducted using sulfuric acid (H$_2$SO$_4$) and hydrogen peroxide (H$_2$O$_2$) as leaching agents [13]. This last reagent was used as an oxidant for iron (Fe$^{2+}$ passed to Fe$^{3+}$). The experiments were developed inside a 400 mL (0.4 dm$^3$) Pyrex beaker in a temperature-controlled shaking bath. The results showed that recoveries of copper and iron close to 99% and 36% were achieved, respectively, using a temperature of 80 °C (353 K), 1.6 M H$_2$SO$_4$ (1.6 mol/dm$^3$), 700 rpm and 10 g (1 × 10$^{-2}$ kg) of solids. In addition, in [13], copper was precipitated as copper sulfate pentahydrate through the use of ethanol, methanol and sulfuric acid (99%, 98% and 73% precipitation, respectively).

Other studies have worked with different indirect analysis methods to obtain the best malachite dissolution parameters, such as by using the Taguchi method (as in Kurşuncu et al. [14]). Other studies have considered leaching malachite in organic solutions such as 5-SSA (5-sulfosalicylic acid) [15].

The studies mentioned above do not show the intrinsic influence of NH$_4$OH on leaching of malachite (only when mixed with other reagents), that is, the effectiveness and the reactions that occur are not clear. Therefore, they fail to obtain the reaction mechanisms of the reagent or the model representing the reaction or its kinetic parameters. In addition, one of the studies had to be carried out with a high reaction temperature (343 K) to obtain maximum malachite dissolution. The present study aims to obtain the malachite dissolution mechanisms in an ammonium system such as NH$_4$OH by analyzing the variables of solution pH, stirring rate, temperature, ammonium hydroxide concentration, solid/liquid ratio and different ammonium reagents. The heterogeneous kinetic model representing the leaching of Cu$_2$CO$_3$(OH)$_2$ will also be obtained along with its kinetic parameters, such as activation energy and reaction order with regard to NH$_4$OH.

## 2. Mechanism of Malachite Dissolution

The dissolution of malachite is given by the chemical process, that is, copper carbonate has a covalent bond and is insoluble to water but soluble in the presence of certain ions in the solution. Künkül [11] also indicated that malachite follows a simple dissolution process, where the oxide/reduction reaction (electron exchange) is not involved. Since it has a low value of the dissociation constant [16], this copper carbonate dissolves according to the reaction (4).

$$Cu_2CO_3(OH)_2 \rightarrow 2Cu^{2+} + CO_3{}^{2-} + 2(OH)^- \qquad k_{sp} = 4.0 \times 10^{-29} \qquad (4)$$

In conditions of high basicity, cupric ions ($Cu^{2+}$) can precipitate due to the formation of copper hydroxide species ($Cu(OH)_2$). However, in the presence of ammonia ions, the solubility of copper species is very high [7,8,17]. In order to corroborate this complex reaction, a speciation diagram of the Cu-$NH_3$ system was constructed at a concentration of 0.005 mol/dm$^3$ copper and a temperature of 343 K (Figure 1). It can be seen that as the concentration of $NH_3$ increases, the amount of copper complexes increases. Thus, copper ammine ($Cu(NH_3)^{2+}$, log k = 3.71), copper bi-ammine ($Cu(NH_3)_2{}^{2+}$, log k = 3.07), copper tri-ammine ($Cu(NH_3)_3{}^{2+}$, log k = 2.54) and copper tetra-ammine ($Cu(NH_3)_4{}^{2+}$, log k = 1.79) are obtained. Copper tetra-ammine is stable from an ammonia concentration of 0.01 mol/dm$^3$ (log{$NH_3$} = −2) and is completely stable for an $NH_3$ concentration of 0.10 mol/dm$^3$ (log{$NH_3$} = −1). The other species of copper tetra-ammine (ammine, bi and tri) are thermodynamically unstable [17].

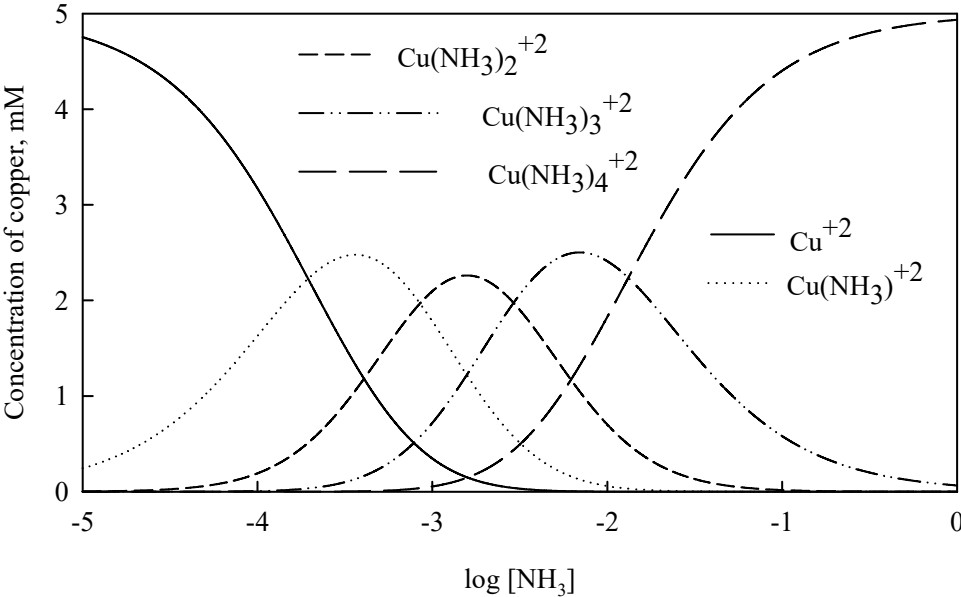

**Figure 1.** Formation of copper and ammonia complexes based on the concentration of $NH_3$.

Ammonium hydroxide, when in contact with water, dissociates (reaction (5)) to form several ionic species, including the hydronium ion ($H_3O^+$). This species has been the main oxidant of several copper oxides [7]. In this case, as the malachite proceeds by a chemical process, $H_3O^+$ would have no chance of reacting with anything besides the ions generated in reaction (4); therefore, it hydrolyzes to produce water. Thus, the overall dissociation reaction of the ammonium hydroxide can be given by the reaction 6.

$$NH_4OH + H_2O \rightarrow NH_3 + H_3O^+ + OH^- \tag{5}$$

$$NH_4OH \rightarrow NH_3 + H_2O \tag{6}$$

Therefore, copper in an ammonium system and base system is stable in the copper tetra-ammine form (shown in Figure 1), as represented by Equation (7).

$$2Cu^{2+} + 8NH_3 \rightarrow 2Cu(NH_3)_4{}^{2+} \tag{7}$$

Thus, the general equation that represents the malachite leaching with ammonium hydroxide is given by Equation (8).

$$Cu_2CO_3(OH)_2 + 8NH_4OH \rightarrow 2Cu(NH_3)_4{}^{2+} + CO_3{}^{2-} + 2(OH)^- + 8H_2O \tag{8}$$

Given how carbonates balance with water [18] Equation (8) can be expressed as:

$$Cu_2CO_3(OH)_2 + 8NH_4OH \rightarrow 2Cu(NH_3)_4{}^{2+} + HCO_3{}^- + 3(OH)^- + 7H_2O \tag{9}$$

Therefore, malachite leaching with ammonium hydroxide alone can be carried out without problems, considering the pH values of the solution.

## 3. Materials and Methods

### 3.1. Malachite Samples

Malachite samples (Sigma Aldrich, Santiago, Chile) were obtained from Sigma Aldrich in the form of very fine powders (less than 5 μm) with a purity of 99.5%. Pelletizing was used for the experiments with different particle sizes. The particles were agglomerated by controlled pressure, creating spheres measuring 12, 24 and 36 μm.

### 3.2. Acid Tests

The experimental development for the acid tests was carried out in a 1 dm$^3$ capacity reactor. A mechanical stirrer with Teflon rod was used. A condensate system added to the reactor served to minimize evaporation. A thermocouple was used to record the temperature of the solution. The amount of malachite used was $1.0 \times 10^{-3}$ kg. The volume of sulfuric acid leaching solution was 0.4 dm$^3$. According to the studied pH value, concentrated sulfuric acid (98% purity) was added. The working temperature was 294 K. After 5400 s of each experiment, the liquid samples were filtered and sent for analysis by atomic absorption spectroscopy (AAS) with a Hitachi Z-8100 Zeeman kit (Hitachi High-Technologies Corporation, Tokyo, Japan).

### 3.3. Ammoniacal Leaching

A batch experiment system was used. The details of the reactor were established in a prior study [17]. Briefly, the equipment comprised a heating blanket, water cooled condenser to minimize evaporation losses, thermocouple and mechanical stirrer. ginstruments came from Hess (Santiago, Chile). The glass reactor had a total volume of 2.0 dm$^3$.

The reactor was subsequently loaded with 1.0 dm$^3$ of ammonium hydroxide leaching solution. In some cases, other ammonium reagents were added: ammonium sulfate (($NH_4)_2SO_4$) (Arquimed, Santiago, Chile) with 99.9% purity from Arquimed, ammonium fluoride ($NH_4F$) with 99.6% purity from AnalaR (AnalaR NORMAPUR$^®$, Santiago, Chile) and ammonium nitrate ($NH_4NO_3$) with 98.0% purity from Vimaroni (Vimaroni, Santiago, Chile). Depending on the experiment, the leaching solution was heated or cooled. A mass of $1.0 \times 10^{-3}$ kg of malachite sample was then added to the reactor. The reaction was initiated, and liquid samples were extracted at regular time intervals for subsequent analysis (AAS). At the end of each experiment, the solution was filtered and the residue was then washed and dried for X-ray diffraction (XRD) analysis using a Bruker diffractometer (Bruker Scientific LLC, Billerica, MA, USA) model D4 Endeavor, operated with Cu radiation and Ni Kβ radiation filter.

## 4. Results and Discussion

### 4.1. Sulfuric Acid System Analysis

Experiments related to obtaining the specific consumption of $H_2SO_4$ were developed from malachite, without the interference of impurities (carbonates). The results of the leaching tests of malachite with sulfuric acid are found in Figure 2. This figure shows the consumption of sulfuric acid per dry ton of malachite used, along with copper recovery depending on the pH of the solution. The pH values were 1.0, 2.0, 2.5, 3.0 and 4.0.

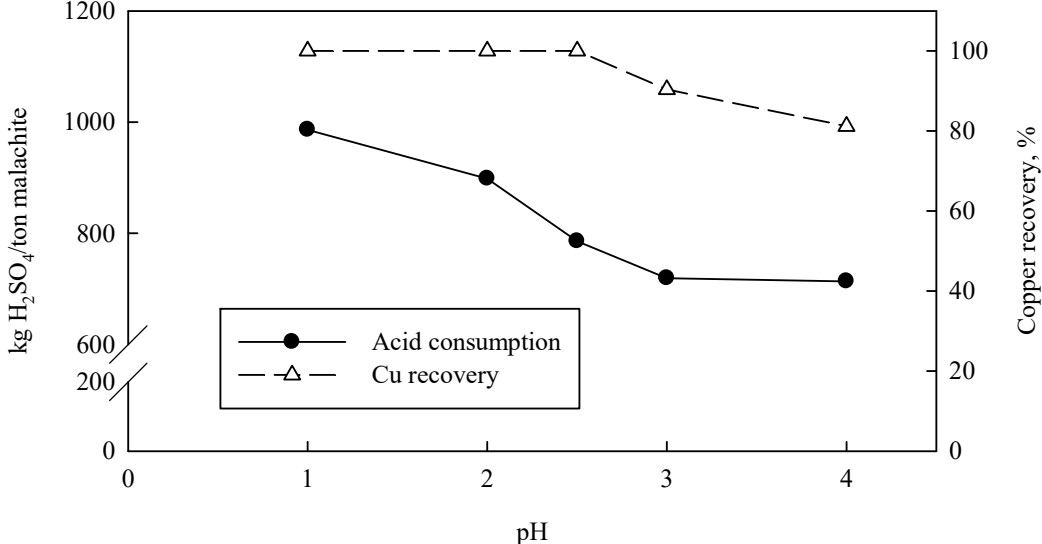

**Figure 2.** Sulfuric acid consumption (kg $H_2SO_4$/ton malachite) and copper recovery (%) depending on the pH of the solution.

It can be seen in this figure that the acid consumption is very high (excessive), reaching values of 986 kg $H_2SO_4$/ton of malachite used for a pH of 1.0. Copper recovery reached up to 99.9%. As the pH increased (became less acidic), the consumption of $H_2SO_4$ and the recovery of copper decreased, reaching values of 713 kg $H_2SO_4$/ton malachite and 81.2% copper, respectively. For efficient acid leaching to be carried out in terms of acid consumption, such consumptions generally must reach values of 50 kg $H_2SO_4$/ton of mineral [1]. With these results shown in Figure 2, acid leaching from malachite could not be carried out. This high consumption may be due to the reaction of malachite with sulfuric acid (reaction (10)) [6].

$$Cu_2CO_3(OH)_2 + 4H^+ = 2Cu^{2+} + CO_2 + 3H_2O \tag{10}$$

Thus, the malachite leaching process using sulfuric acid at room temperature would not be advisable to perform. Therefore, it becomes attractive to be able to develop the leaching of malachite in a basic system, since low sulfuric acid consumption and increased copper recovery are promoted. The leaching solution chosen was ammonium hydroxide.

### 4.2. Zone of Malachite Dissolution in Ammoniacal System

In order to study the zone of malachite dissolution in an ammonium system, tests were conducted at different pH values (6.0, 10.5 and 13.0), an ammonium hydroxide concentration of 0.1 mol/dm$^3$, a temperature of 298 K and stirring at 500 rpm. The solid/liquid ratio used was 1/1000, and the results are shown in Figure 3 as copper recovery as a function of solution pH.

It can be seen that at a pH of 10.5, the copper recovery reached a value of 63.9%. This level of recovery was mainly due to the formation of copper tetra-ammine, as shown in Figure 1. At a pH of 6.0 or 13.0, the copper recovery levels reached only 1.9%. This behavior may be due to copper precipitation in the form of CuO, based on the thermodynamic study shown in the Cu-NH$_3$-H$_2$O stability diagram (Figure 1). In order to corroborate the malachite dissolution mechanisms for different pH values, the solid samples obtained after 7200 s in tests carried out at pH levels of 10.5 and 13.0 were sent for XRD analysis. The results are shown in Figure 4.

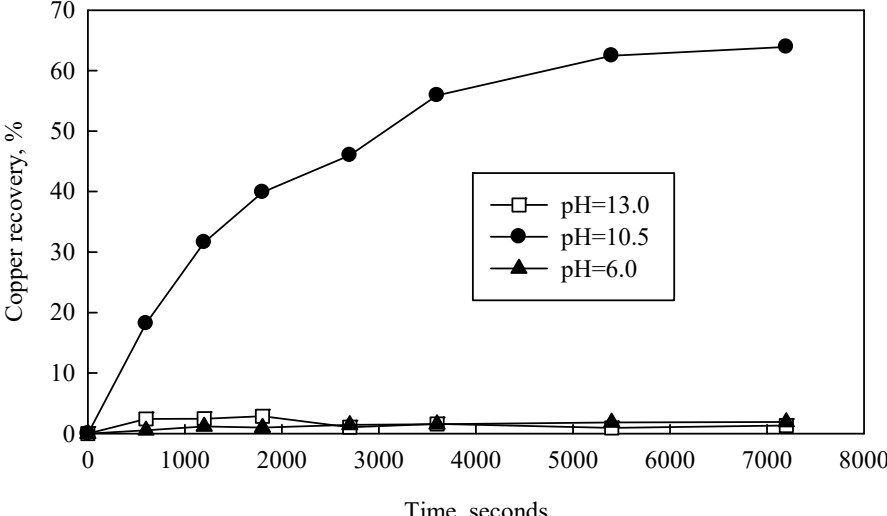

**Figure 3.** The effect of the solution pH on the dissolution of $Cu_2CO_3(OH)_2$. Working conditions: $NH_4OH = 0.1$ mol/dm$^3$, temperature = 298 K, particle size = 5 µm, stirring rate = 500 rpm and S/L ratio = 1:1000.

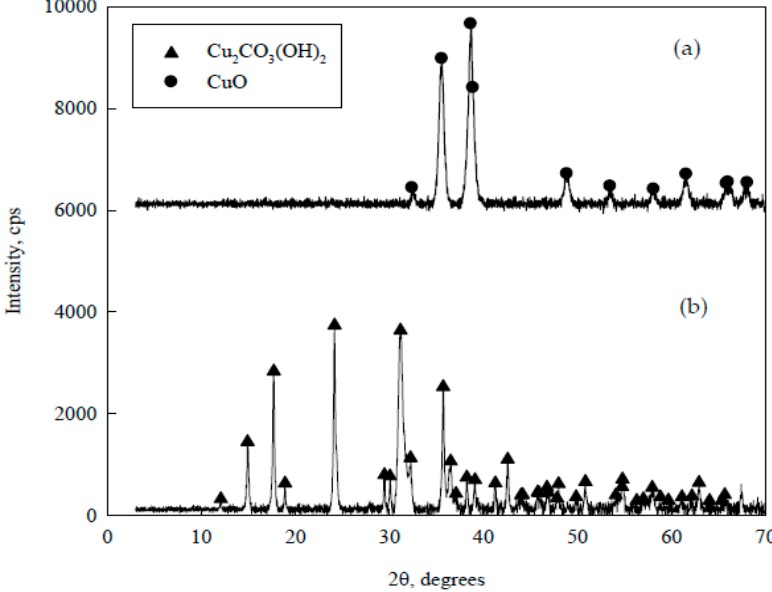

**Figure 4.** XRD analysis of solid samples: (**a**) samples obtained at a pH of 10.5; (**b**) samples obtained at a pH of 13.0.

As can be seen in Figure 4a, the peaks indicate malachite without the presence of any other compound associated with copper or ammonia. Therefore, the dissolution mechanism of $Cu_2CO_3(OH)_2$ should be that shown in Equation (8). In Figure 4b, no malachite is seen, though there are important peaks for copper oxide, such as tenorite. To corroborate this formation of copper oxide, a predominance diagram was built for the Cu-NH$_3$-H$_2$O system for three different temperatures (278, 298 and 313 K). The thermodynamic data were taken from the database of the HSC Chemistry 6.0 program [19]. The copper and ammonium concentrations were 0.0043 and 0.1 mol/dm$^3$, respectively. The potential values used were above 0.4 V. The diagram is presented in Figure 5. It can be seen that the cupric ion is stable at pH less than 4.8. However, it becomes stable again in the pH range from 9.2 to 12.5. This regained stability is due to complexing with the ammonium ion, generating copper tetra-ammine $(Cu(NH_3)_4^{2+})$. This stability is seen for all potential values. The other copper complexes (other

ammines) are not considered due to their thermodynamic instability. In addition, by increasing the temperature, the range of stability of the copper tetra-ammine moves to more acidic pH values, from a pH of 9.8 (278 K) to 9.0 (313 K). Outside these pH ranges, the copper oxidizes and precipitates as cuprite. This happens at all temperatures.

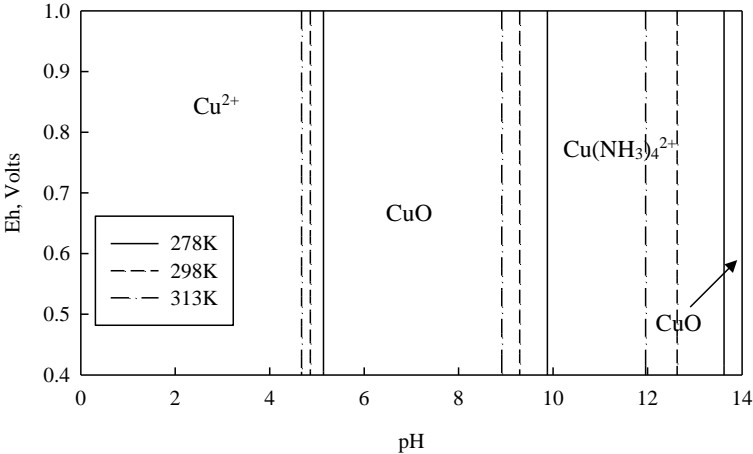

**Figure 5.** Diagram of Cu-NH$_3$-H$_2$O medium stability at a copper concentration of 0.0043 mol/dm$^3$ and ammonia concentration of 0.1 mol/dm$^3$. The solid line corresponds to the equilibrium arising at 278 K, the dotted line is equilibrium at 298 K and the dashed line is equilibrium at 313 K.

Therefore, and according to what is seen in Figure 3, at the more basic pH (13.0), copper recovery was low due to the formation of this oxide (CuO), as was posited thermodynamically previously (Figure 5). The copper must be present in solution as copper tetra-ammine, but in highly base conditions it is precipitated to form tenorite, leading to the low level of recovery of the metal of interest (cupric ions).

On the other hand, the curves shown in Figure 1 show that there would be only one form of chemical reaction of malachite with NH$_4$OH, which could be represented by a single heterogeneous kinetic model (as will be seen later). Oudenne and Olson [20] studied the kinetics of leaching from malachite in an ammonium carbonate solution, where they found that there were two reaction stages: stage I, where a 10% reaction was obtained (quickly) but then became slow, and then stage II where 90% reaction is obtained (total malachite dissolution). Oudenne pointed out that in stage I the reaction became very slow due to a blockage in the surface generated by an intermediate compound formed in the reaction, Cu(OH)$_2$. This compound can be dissolved by the intervention of the hydronium ion. In our case, the generation of the intermediate compound was not evident; therefore, the dissolution of the malachite was always carried out by the chemical process (reaction (9)).

Based on the results observed for the effect of pH, the subsequent experiments were all carried out at a pH of 10.5.

### 4.3. Evaluation of Stirring Rate

The stirring of the leaching solution was assessed in the range of 200 to 600 rpm, including a test without stirring (0 rpm). The tests were carried out in a solution of 0.1 mol/dm$^3$ NH$_4$OH, at 298 K, with a solid/liquid ratio of 1/1000. The results are shown in Figure 6.

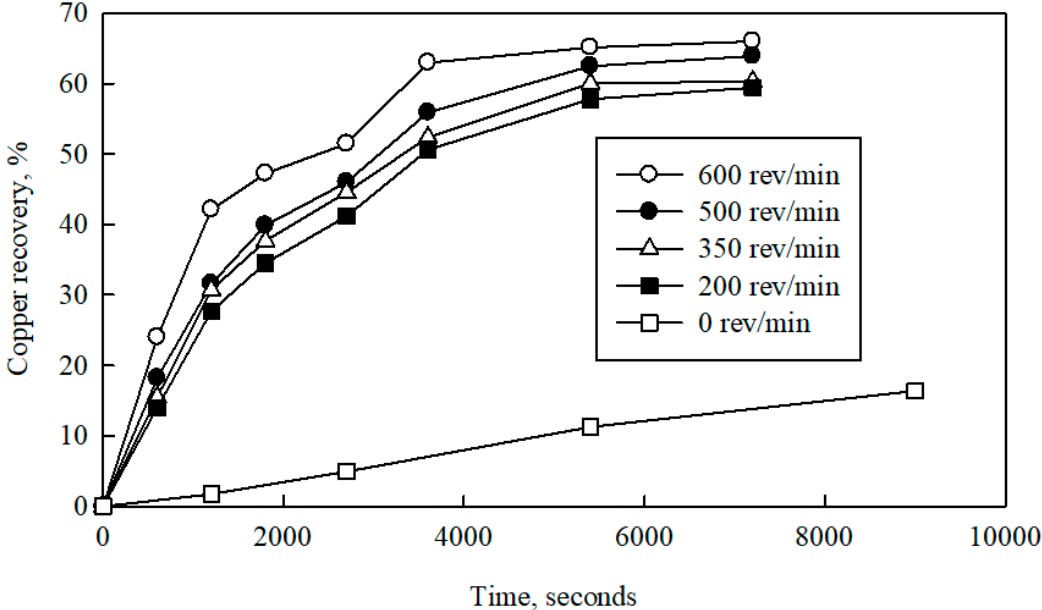

**Figure 6.** The effect of stirring rate on the malachite dissolution. Working conditions: $NH_4OH = 0.1$ mol/dm$^3$, temperature = 298 K, pH = 10.5, particle size = 5 μm and S/L ratio = 1:1000.

The figure shows that copper recovery increases as the stirring rate increases. Thus, for a stirring rate of 200 rpm, recovery of 64.3% was obtained after 7200 s; when increasing the stirring to 500 rpm, the Cu recovery reached a value of 71.0% for the same experiment time. For higher stirring rates, the copper recovery remained similar. This is due to the phenomenon of mass transference no longer playing a significant role at higher stirring rates. Therefore, all subsequent experiments were conducted at a stirring rate of 500 rpm to ensure that the stirring rate was not affected by mass transfer.

It should be noted that the recovery rate increased with time during the experiment conducted without stirring (0 rpm), reaching a maximum copper recovery of 16.4% after 9000 s.

### 4.4. Temperature Analysis

The effect of temperature on the dissolution rate of $Cu_2CO_3(OH)_2$ was assessed. The range of temperatures tested was 278 to 313 K. The working conditions used were 0.1 mol/dm$^3$ $NH_4OH$ and a solid/liquid ratio of 1/1000. As can be seen in Figure 7, there was a significant effect on the early dissolution times, with this effect decreasing after 3600 s (except for the curve generated at 313 K). Maximum dissolution reached a value close to 72% for a temperature of 313 K. It can also be seen that at the lower temperature (278 K), which is close to the freezing point of water (273 K), significant copper recovery was also obtained (56.1%) after 7200 s.

The differences may be due to the changes in the kinetic constants involved in the malachite dissolution processes.

It should also be considered that ammonia volatilizes slowly in equilibrium with ammonia in solution (ammonia dissociation constant = $1.77 \times 10^{-5}$ at 298 K) increasing with temperature, thus decreasing its concentration in the solution as the dissolution time elapses. This dissociation of Cu-NH$_3$ will follow the series set forth by the diagram in Figure 1, beginning with $Cu(NH_3)_4^{2+}$ until reaching the form of $Cu^{2+}$, as the concentration of NH$_3$ decreases (in solution). Then, the cupric ion under conditions of high alkalinity would precipitate as an oxide, thus decreasing the concentration of copper in the solution.

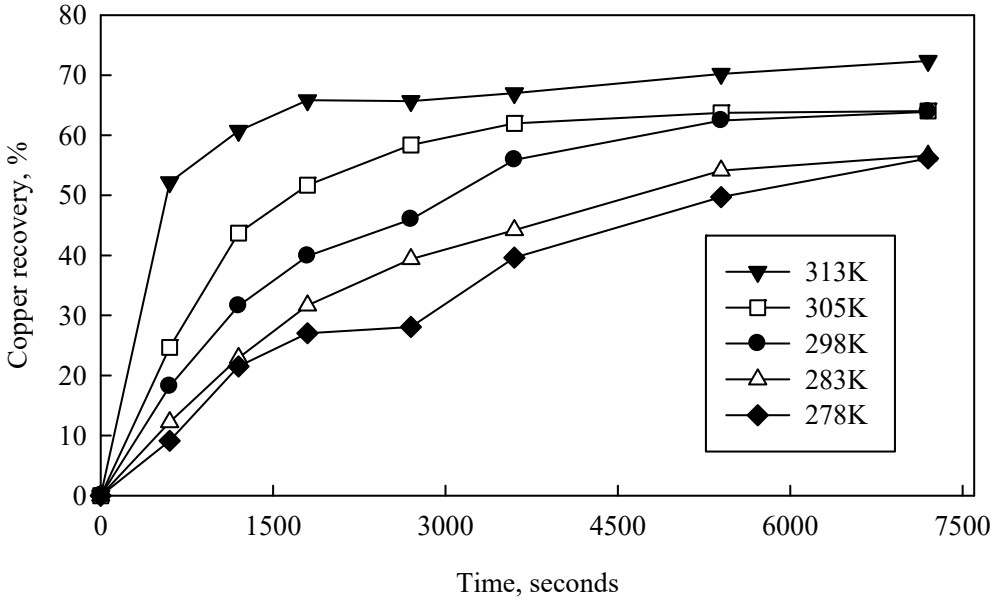

**Figure 7.** Analysis of the effect of temperature on the rate of malachite dissolution. Working conditions: $NH_4OH$ = 0.1 mol/dm$^3$, pH = 10.5, particle size = 5 μm, stirring rate = 500 rpm and S/L ratio = 1:1000.

### 4.5. Effect of NH$_4$OH Concentration

The study of the ammonium medium was carried out at 298 K with a solid/liquid ratio of 1/1000. The concentration values ranged from 0.01 to 0.2 mol/dm$^3$ $NH_4OH$. The results are shown in Figure 8. Based on Equation (9) and using stoichiometry, the minimum required $NH_4OH$ concentration to extract copper from malachite was found to be 0.036 mol/dm$^3$. It can be seen in Figure 8 that no copper was recovered from the malachite when using a concentration of 0.01 mol/dm$^3$. However, when increasing the concentration to 0.05 mol/dm$^3$ (equal to or greater than the stoichiometric level), copper recovery reached a value of 19.3% after 7200 s. For the maximum concentration of ammonium hydroxide (0.2 mol/dm$^3$), copper recovery obtained reached a value of 84.1% for the same experiment time.

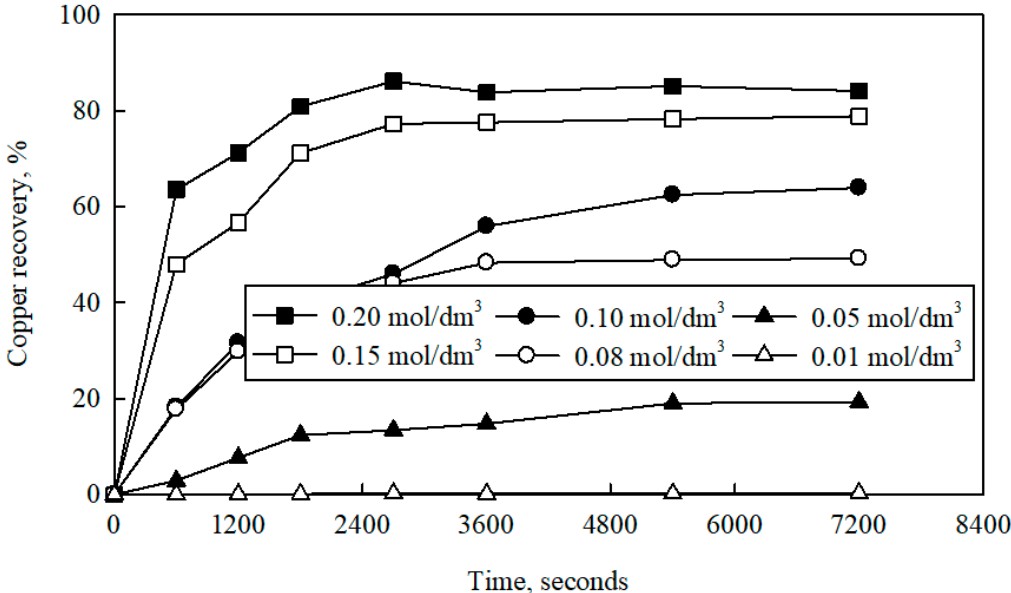

**Figure 8.** The effect of ammonium hydroxide concentration. Working conditions: temperature = 298 K, particle size = 5 μm, stirring rate = 500 rpm, pH = 10.5 and S/L ratio = 1:1000.

### 4.6. Evaluation of Particle Size

The effect of the particle size of the malachite on its leaching rate was also evaluated. Four tests were carried out with different particle sizes: 5, 12, 24 and 36 μm. The temperature and $NH_4OH$ concentration were set at 298 K and 0.1 mol/dm$^3$, respectively. Figure 9 shows the copper recovery as a function of time for the different particle sizes.

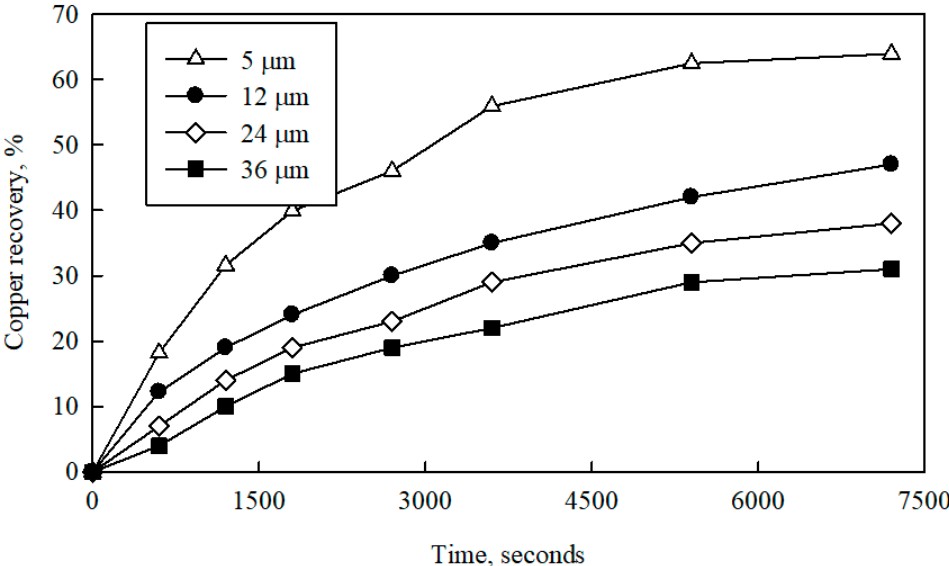

**Figure 9.** The effect of average particle size on copper recovery. Working conditions: $NH_4OH = 0.1$ mol/dm$^3$, temperature = 298 K, pH = 10.5, stirring rate = 500 rpm and S/L ratio = 1:1000.

It can be seen in the figure that there is increased copper recovery as the particle size decreases. Thus, for a particle size of 36 μm (and a time of 7200 s), copper recovery of 31.0% was obtained; at a particle size close to 7 times smaller, copper recovery increased almost 2-fold. This is mainly because the smaller particle size increases the area of the reaction interface between the $Cu_2CO_3(OH)_2$ and ammonia molecules.

### 4.7. Analysis of the Solid/Liquid Ratio

In order to evaluate the solid/liquid ratio, tests were carried out lasting 3600 s. The working conditions were a temperature of 298 K and $NH_4OH$ concentration of 0.1 mol/dm$^3$. Different ammonium solution volumes were used, ranging from 0.1 to 0.8 dm$^3$, maintaining a constant mass of malachite of $1 \times 10^{-3}$ kg. Figure 10 summarizes the copper extraction results as a function of the solid/liquid ratio.

The main objective was to obtain the maximum possible level of copper recovery with the lowest solution volume. This was achieved using a ratio of 0.6 dm$^3$/kg, which reported a copper recovery of 72.0% due to a more efficient reaction medium between the diffusion of $NH_3$ and $Cu_2CO_3(OH)_2$. It can also be seen that Cu recovery reached only 7.0% when using the lowest ratio (0.1 dm$^3$/kg).

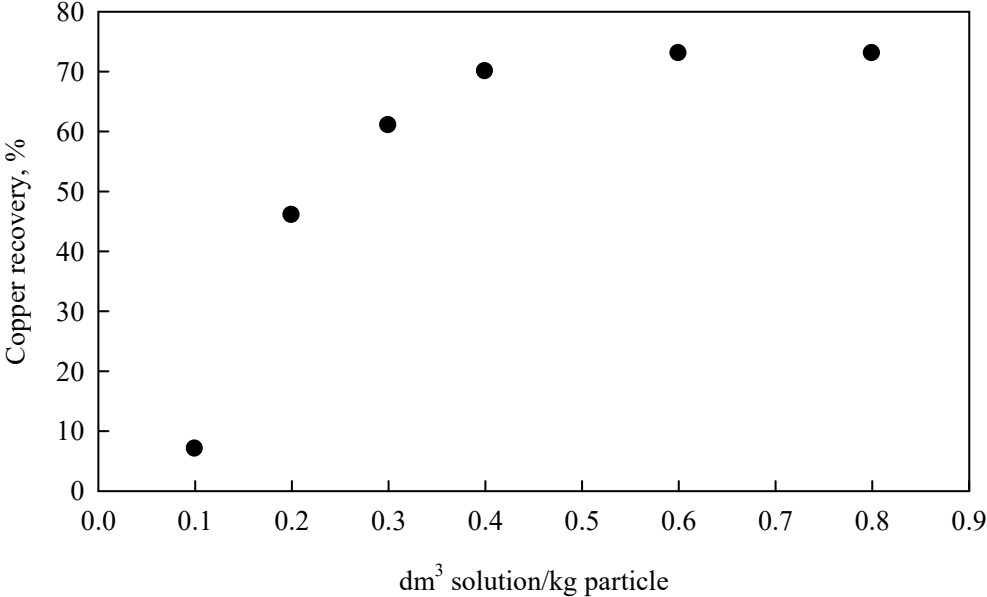

**Figure 10.** Evaluation of the effect of the solid/liquid ratio on copper recovery. Working conditions: $NH_4OH$ = 0.1 mol/dm$^3$, temperature = 298 K, particle size = 5 μm, pH = 10.5, stirring rate = 500 rpm and experiment time = 3600 s.

### 4.8. Effect of Different Ammonium Reagents

Figure 11 shows the results for copper recovery as a function of time for different ammonium reagents: $(NH_4)_2SO_4$, $NH_4F$, $NH_4NO_3$ and $NH_4OH$. There is a clear positive effect on the dissolution of $Cu_2CO_3(OH)_2$ for the four reagents used, obtaining values close to 24.2% after 7200 s for the ammonium nitrate. However, when using ammonium fluoride and ammonium sulfate, the copper recovery values reached only 9.2% and 80.9%, respectively, for the same experiment time. For these two reagents ($NH_4F$ and $(NH_4)_2SO_4$), the copper recovery becomes extremely slow after 20 min.

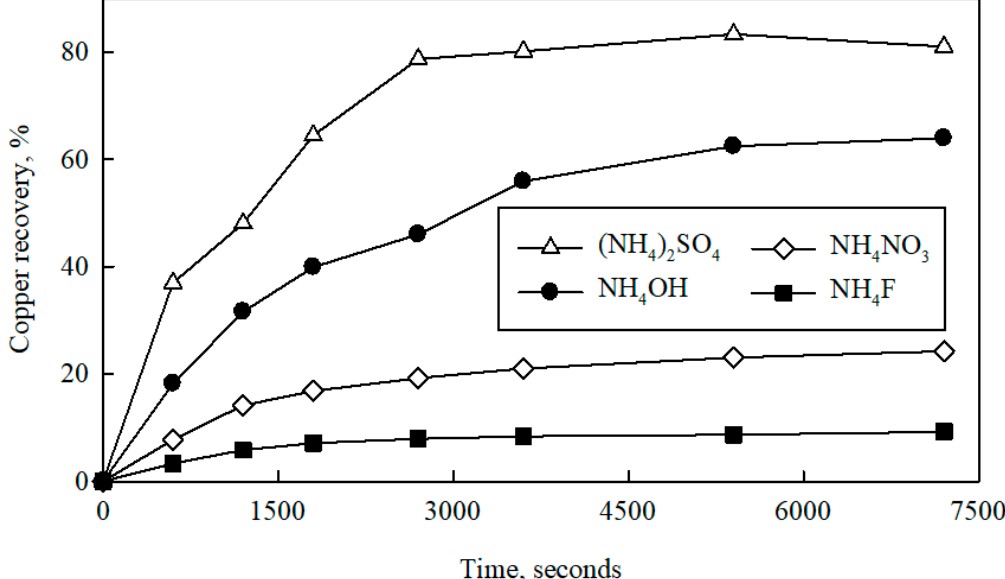

**Figure 11.** Evaluation of the influence of different ammonium reagents on the malachite leaching rate. Working conditions: $NH_4OH$ = 0.1 mol/dm$^3$, temperature = 298 K, particle size = 5 μm, pH = 10.5, stirring rate = 500 rpm and S/L ratio = 1:1000.

It should be noted that the pH values remained constant at 10.5. Therefore, due to the lack of prior studies of oxide leaching using ammonium reagents, the use of other ammonium reagents requires additional research to find the maximum possible dissolution of malachite.

The copper generated in the dissolution of malachite can be recovered by a process of solvent extraction (SX) with electrowinning (EW). Some research studies have used SX to recover copper and ammonia using sterically hindered β-diketone [21], to recover copper using LIX 54 [22] or with the use of liquid membranes using LIX 84I [23]. Therefore, copper concentrated by SX can be obtained through electrowinning as metallic copper (with a cathode of high purity) or precipitated as copper sulfate.

### 4.9. Dissolution Kinetics

According to Figure 7, the effect of temperature on the rate of malachite dissolution was not significant. This suggests that the malachite dissolution is governed by a process of diffusion in a porous layer due to the dissolution of the particle as it cracks. This model represents particles that begin to react as nonporous but become porous during the reaction, i.e., the original solid cracks and splits to form a porous structure resembling a granular material, with each grain reacting through a decreasing core mechanism (shrinking core model). Therefore, the reaction rate follows a shrinking core model in which diffusion is controlled by the porous layer with an initial radius for a constant reagent concentration, expressed as in the following equation [24]:

$$1 - \frac{2}{3}\alpha - (1 - \alpha)^{\frac{2}{3}} = k_{app}\, t \tag{11}$$

In this equation, the converted fraction, $\alpha$, represents the conversion of the malachite at time t. The apparent reaction rate constant is represented by the following expression:

$$k_{app} = k_o \frac{b\,[NH_4OH]^n}{r_o^2} e^{-E_a/RT} \tag{12}$$

In this expression, $[NH_4OH]$ and n are the concentration and order to the reaction with regard to the ammonium hydroxide concentration, respectively; $k_o$ is the intrinsic reaction rate constant; $b$ is the stoichiometric constant given by Equation (9), which relates the molarity between the ammonium hydroxide and the malachite; and $r_o$ is the initial radius of the particles.

Using the experimental data shown in Figure 7, a graph was built to represent the diffusion model through the porous layer as a function of time for the temperature range of 278 to 313 K. Figure 12 indicates the results; it can be seen that there is a good linear fit of the experimental data, with the regression coefficients ($R^2$) being close to 0.96 for the entire temperature range. These high values of $R^2$ validate the Kinetic Equation (11). The apparent reaction rate constants (the gradient of each straight line) are presented in Table 1.

**Table 1.** Value of each apparent kinetics constant for the five temperatures studied.

| T [°C (K)] | 1000/T (1/K) | $k_{app}$, 1/s |
|:---:|:---:|:---:|
| 5 (278) | 3.5971 | $6.3 \times 10^{-6}$ |
| 10 (283) | 3.5336 | $7.9 \times 10^{-6}$ |
| 25 (298) | 3.3557 | $12.0 \times 10^{-6}$ |
| 32 (305) | 3.2787 | $18.8 \times 10^{-6}$ |
| 40 (313) | 3.1949 | $44.7 \times 10^{-6}$ |

The experimental data in Figure 8 for the $NH_4OH$ concentration range of 0.08 to 0.2 mol/dm$^3$ were also used to build a graph of the diffusion model in the porous layer as a function of time (Figure 13). It can be seen in this figure that a good fit is obtained for the experimental data for all the straight lines generated ($R^2 > 0.92$).

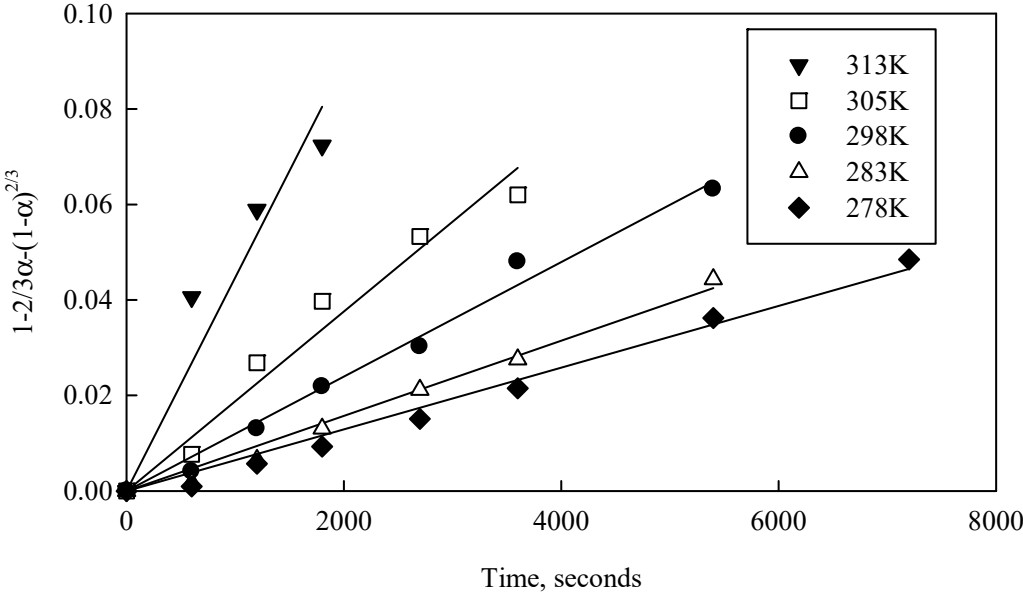

**Figure 12.** Analysis of malachite leaching kinetics as a function of temperature. The working conditions are the same as those in Figure 7.

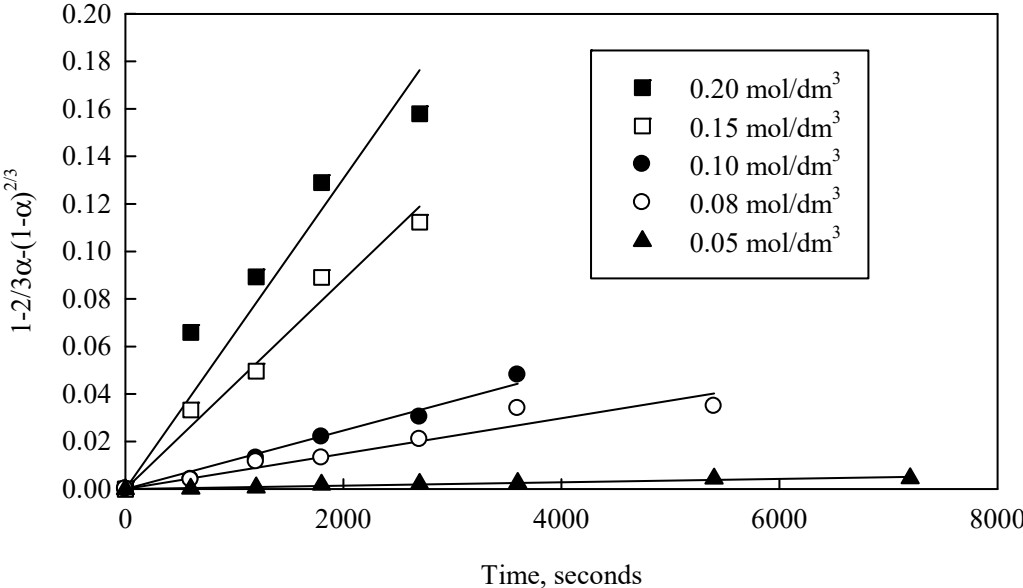

**Figure 13.** Evaluation of the kinetic model for different ammonium hydroxide concentrations. Working conditions are the same as those in Figure 8.

The values of $k_{app}$ were used to build a graph of $\ln(k_{app})$ as a function of $\ln([NH_4OH])$, as shown in Figure 14. This figure shows a good linear fit, with $R^2$ values reaching 0.95. The gradient of the straight line corresponds to the value of the reaction order (n) for the specific ammonium hydroxide concentration. Therefore, the reaction order calculated for the malachite dissolution is 3.2.

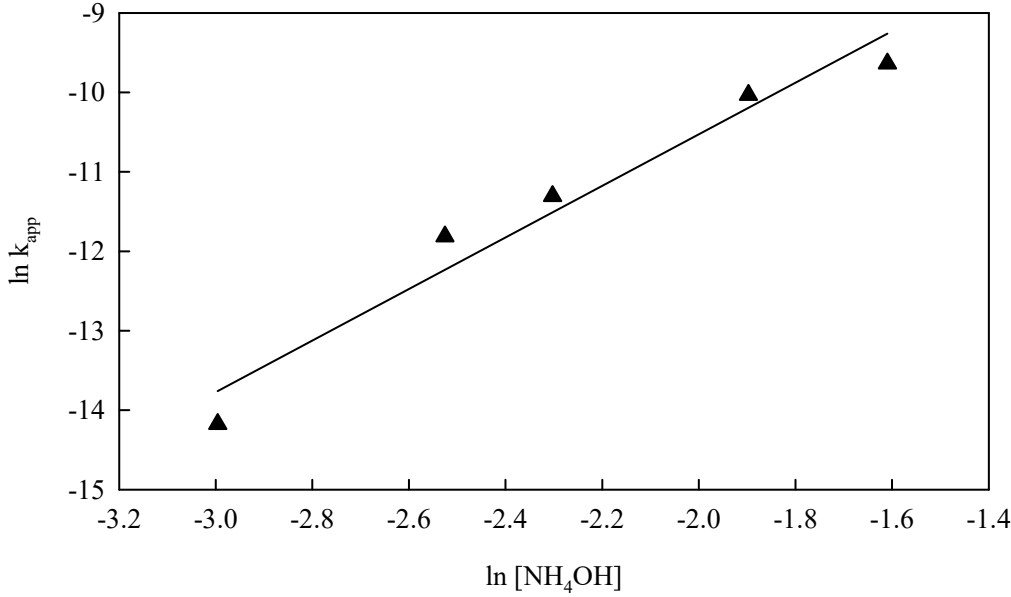

**Figure 14.** Reaction order calculated with regard to the concentration of $NH_4OH$.

For a kinetic model that is controlled by diffusion through a porous layer, the apparent constant values should vary linearly with the inverse square of the initial particle radius, as seen in Equation (12). In order to verify this, the particle size data (Figure 9) were entered into Equation (12), generating the graph shown in Figure 15.

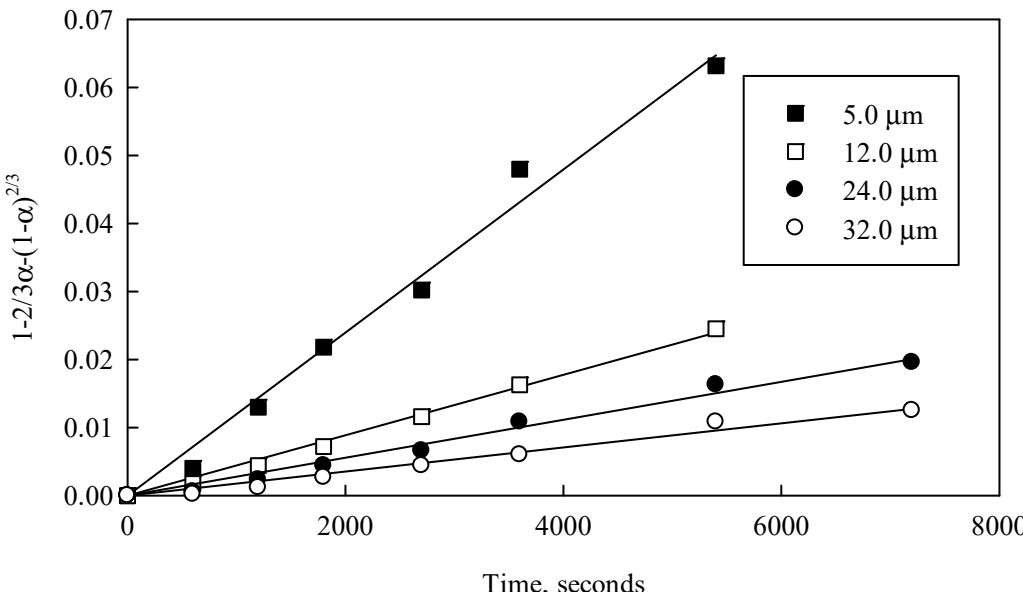

**Figure 15.** Malachite leaching kinetics for different average particle sizes. Working conditions are the same as those in Figure 7.

It can be seen in Figure 15 that a good correlation was obtained ($R^2$ close to 0.98), validating Equation (11) for the diffusion model in a porous layer generated postleaching. The values of $k_{app}$ obtained from Figure 15 were graphed in Figure 16 as a function of the inverse square of the initial radius, as shown in Equation (12). The linear dependence of the data shown in Figure 16 ($R^2 > 0.99$) therefore validates the kinetic model used in the present study.

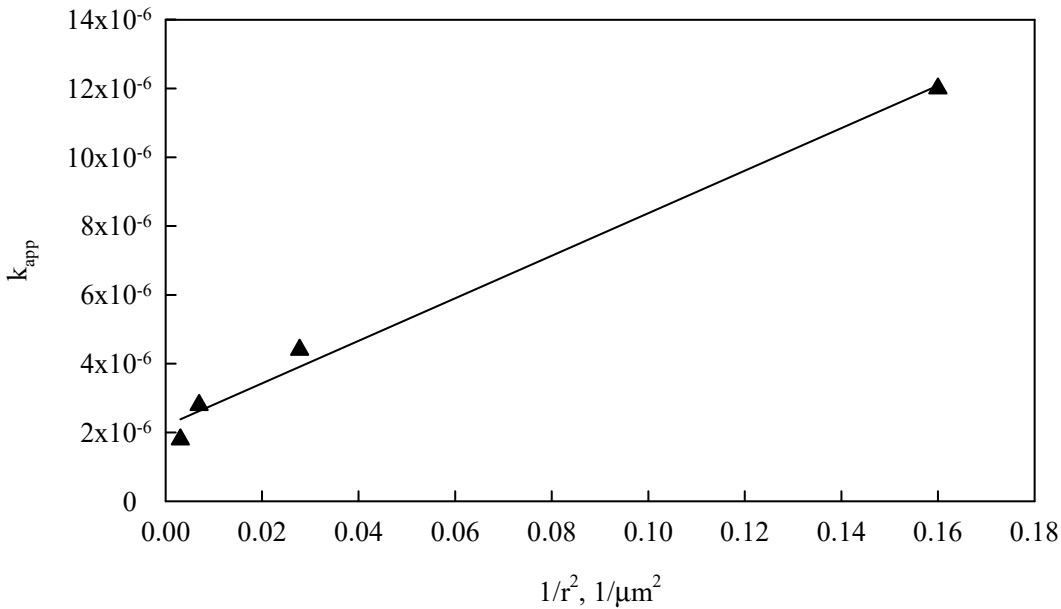

**Figure 16.** Evaluation of the inverse of the average particle size.

In order to calculate the activation energy ($E_a$), the apparent reaction rate constants obtained in Table 1 were used along with the ammonium hydroxide concentration and reaction order values of 0.1 mol/dm$^3$ and 3.2, respectively. The value of *b* was 1/8, according to the stoichiometry in Equation (9). These values were substituted into Equation (12). Table 2 shows the results of the intrinsic reaction rate constant values as a function of the temperature range used in the study.

**Table 2.** Intrinsic reaction rate constants for the malachite leaching in NH$_4$OH.

| T, K | $k_o$, 1/s µm$^2$ 1/(mol/dm$^3$)$^{3.2}$ |
|------|------------------------------------------|
| 278  | $49.92 \times 10^{-2}$ |
| 283  | $62.60 \times 10^{-2}$ |
| 298  | $95.09 \times 10^{-2}$ |
| 305  | $148.98 \times 10^{-2}$ |
| 313  | $354.22 \times 10^{-2}$ |

An Arrhenius plot was then built using the values of $k_o$ for the temperature range in study. Figure 17 shows a good linear fit ($R^2 = 0.90$) for the temperature dependence with regard to the kinetics constants. The activation energy was calculated as 36.1 kJ/mol for the temperature range of 278 to 313 K. This value is typical for a diffusion model through a porous layer. Therefore, the kinetic equation representing the malachite leaching in an ammonium system (NH$_4$OH) is that shown in expression (13):

$$1 - \frac{2}{3}\alpha - (1-\alpha)^{\frac{2}{3}} = 2.85 \times 10^6 \frac{\frac{1}{8}[\text{NH}_4\text{OH}]^{3.2}}{r_o^2} e^{-36.1/RT} \, t \tag{13}$$

where *R* is the gas constant and is equal to 8.314 J/mol/K, [NH$_4$OH] is in mol/dm$^3$, $r_o$ is in µm, t is in seconds, *T* is in Kelvin and $k_o$ equals $2.85 \times 10^6$ 1/s µm$^2$ 1/(mol/dm$^3$)$^{3.2}$.

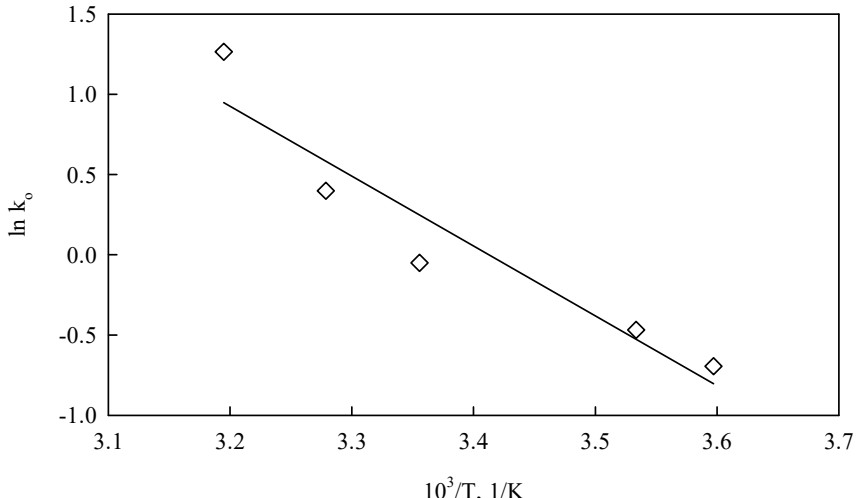

**Figure 17.** Arrhenius plot for the temperature range of 278 to 313 K.

## 5. Conclusions

This research aimed to obtain the reaction mechanism and analyze the kinetics of malachite leaching with the use of ammonium hydroxide at different temperatures. The innovative use of this leaching solution was done mainly because the principle component of malachite is carbonate, which consumes large amounts of sulfuric acid (the most commonly used leaching compound). For our study, the excessive value of acid consumption by malachite was 986 kg $H_2SO_4$/ton of malachite. The use of $NH_4OH$ avoids the need to use $H_2SO_4$, leading to useful metal (copper) extraction.

The results obtained are promising, showing copper recovery above 82% (ammoniacal system). Increasing the temperature and ammonium hydroxide concentration led to increased copper recovery, while decreasing the particle size also caused an increase in the recovery rate. The pH of the solution was also a significant factor in the malachite leaching process.

Malachite dissolution is governed by a process of diffusion in a porous layer due to the dissolution of the particle as it cracks, i.e., the original solid cracks and splits to form a porous structure resembling a granular material, with each grain reacting through a decreasing core mechanism.

**Author Contributions:** Conceptualization, A.A.; formal analysis, O.J.; funding acquisition, A.A.; investigation, A.A., O.J.; methodology, J.P.; validation, J.P., A.A., O.J.; writing—original draft, A.A., O.J. authors have read and agreed to the published version of the manuscript.

**Funding:** The APC was financed by the Office of the Vicerrectoría de Investigación y Estudios Avanzados of the Pontificia Universidad Católica de Valparaíso.

**Conflicts of Interest:** The authors declare no conflict of interest.

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
