# Peer review of "Mechanism and Kinetics of Malachite Dissolution in an NH4OH System"

_metals, doi:10.3390/met10060833_

Round 1
Reviewer 1 Report
This paper discusses the effects of the process parameters of pH, stirring rate, temperature, concentration, etc. on leaching Cu2+ from Cu2CO3(OH)2.
The reviewer thinks that important information is missing to discuss the mechanism and he recommends Major revision for this paper.
The comments and inquiries are as follows.
1) The authors describe "ammonium hydroxide", but it should more typically be expressed as ammonia solution, because the ionized species of NH4+ is not major according to the equilibrium constant of the reaction of NH3 with H2O.
The details of the reagents such as purities and makers should be shown in 3. Materials and methods for "ammonium hydroxide", ammonium sulphate, fluoride, and nitrate, even though they were described in the authors’ previous paper.
2) The reviewer does not understand how the authors adjusted the pH independent of the NH3 aq. concentration. Please add the explanation.
The reviewer thinks that the pH will affect the equilibrium shown in Fig. 1. The pH will depend on the concentration of the ammonia solution and the alkaline condition with lower NH3 concentrations will lead to the formation of Cu(OH)2. The absence of the copper hydroxide in Fig. 1 appears to be strange. Please explain the calculation details for Fig. 1.
3) The authors should carefully discuss the results of Fig. 12 and 13. The fitting of the linear approximation is rather poor especially for higher temperatures and concentrations.
4) "copper amine" should be "copper ammine" (P. 4, L. 149 and other places).
"NH3OH" should be "NH4OH" in Eq. (1).
Well, the authors proposed a somewhat complicated reaction in Eq. (1) for dissolution of CuO in ammonia solution. However, it may more simply be expressed as
CuO + 4NH3 + H2O → [Cu(NH3)4]2+ + 2OH-
The reviewer does not understand why the authors concluded the reactions shown in Eqs. (1) and (2). Please add brief explanation by quoting their previous works.
Author Response
Dear Reviewer,
Thank you very much for helping us to generate and improve this research manuscript. It is very important for us to have valuable comments from an expert in the field. Below we send our changes and/or suggestions: (in blue color):
1) The authors describe "ammonium hydroxide", but it should more typically be expressed as ammonia solution, because the ionized species of NH4+ is not major according to the equilibrium constant of the reaction of NH3 with H2O.
The details of the reagents such as purities and makers should be shown in 3. Materials and methods for "ammonium hydroxide", ammonium sulphate, fluoride, and nitrate, even though they were described in the authors’ previous paper.
The referee is right. However, the authors have maintained the description "ammonium hydroxide" in previous articles. Thus, the metallurgical / chemical audience can follow the research work and link each one more easily. Thank you very much for detail.
Regarding the details of the agents, these were added in the text. Thank you.
2) The reviewer does not understand how the authors adjusted the pH independent of the NH3 aq. concentration. Please add the explanation.
The reviewer thinks that the pH will affect the equilibrium shown in Fig. 1. The pH will depend on the concentration of the ammonia solution and the alkaline condition with lower NH3 concentrations will lead to the formation of Cu(OH)2. The absence of the copper hydroxide in Fig. 1 appears to be strange. Please explain the calculation details for Fig. 1.
What happened was that by working with the given ammonium hydroxide concentration, the working pH values were very close. Then, these values could generate the copper hydroxides, as long as ammonium ions were not present. However, as shown in Figure 1, there is always ammonium compound, in order to carry out the reaction of copper ions, thus avoiding the formation of copper hydroxides. Thank you very much for improving our manuscript..
3) The authors should carefully discuss the results of Fig. 12 and 13. The fitting of the linear approximation is rather poor especially for higher temperatures and concentrations.
The referee is right. At a high temperature (313K) as well as at a high concentration (0.20 M) there are some points outside the trend line. This occurred due to the extreme work carried out, generating various difficulties in keeping the development of the experiment intact. However, the data was representative. We believe that basically it would not be necessary to give a deep explanation about these difficulties. Thanks for the gesture.
4) "copper amine" should be "copper ammine" (P. 4, L. 149 and other places).
"NH3OH" should be "NH4OH" in Eq. (1).
Well, the authors proposed a somewhat complicated reaction in Eq. (1) for dissolution of CuO in ammonia solution. However, it may more simply be expressed as
CuO + 4NH3 + H2O → [Cu(NH3)4]2+ + 2OH-
The reviewer does not understand why the authors concluded the reactions shown in Eqs. (1) and (2). Please add brief explanation by quoting their previous works.
Copper ammine was added as well as all ammines.
NH4OH was changed
We cannot change equation (1) since it was taken from another work. Thank you very much for your suggestions and the time dedicated to improving our manuscript. Our sincere thanks.

Reviewer 2 Report
The article concerns a current and very important topic. The subject of malachite dissolution in acid and alkaline solutions is important from the point of view of process efficiency. The introduction in a transparent manner introduces the subject of the article and review of the conducted research. The adopted research methodology is correct. Test results are presented graphically and tabularly, and briefly discussed. The language of the article leaves much to be desired, some sentences need to be redefined. In addition, in Chapter 3 it is necessary to present the research methodology in the form of a diagram to better understand the order of activities.
There are some remarks that should be corrected before eventual publishing:
- Reaction (1), (2) and (3) are not balanced (wrongly written)
- Line 74 ammoniacal system, line 75 ammonium system – please unify (decide to one of this form), in the rest of the article appears several more times
- Temperature once in K once in oC, sometimes in this both, there is suggestion to use only K, the same concerns concentration, mass and time in the whole article (using this two unit is chaotically)
- Line 74 – chapter needs correction (style)
- Line 95 (stirring rate – rev/min or rpm
- Line 103 – malachite solution? Or copper solution
- Line 116 – Fe2+ passed to Fe3+
- Line 111 Arzutug [12]
- Line 120 – in this work [13]
- Line 124 – taguchi method in Kursuncu [14]
- Line 126-127 – sentence should be rearranged
- Line 137 – 138 – sentence not clear, should be rearranged
- Line 145 - sentence not clear, should be rearranged
- 1 – concentration of cooper or coper ions
- Line 176-180 – not clear – should be rearranged
- Line 187-188 – not clear – should be rearranged
- Line 197 - not clear – should be rearranged
- Line 210 - not clear – should be rearranged
- Line 218 chapter 3.3. the same mark as in pkt 2 (ammoniacal or ammonium)
- In chapter 3 – there should be scheme of research – it will improve the visibility of research methodology
- In Fig. 3 time, second
- Line 325 – Figure 1 or Figure 5?
- Line 389-391 - not clear – should be rearranged
- Figure 5 – Cu(NH3)42 – there is need for +
- Line 430 – 432 - not clear – should be rearranged
- Figure 6 , 7, 8, 9, 11 - black boxes appear in the drawings, what they mean, if anything, they need to be removed
- Line 543-544 – not clear – wrongly formulated
- Figure 10 - a better solution would be to present the results in the form of a block diagram
- Line 705 – not clear
- Figure 13 - reference is made to Fig. 6 while in line 731, Figure 8
Author Response
Thank you very much for helping us to generate and improve this research manuscript. It is very important for us to have valuable comments from an expert in the field. Below we send our changes and/or suggestions: (in blue color)
There are some remarks that should be corrected before eventual publishing:
- Reaction (1), (2) and (3) are not balanced (wrongly written)
Reactions were corrected. Thanks a lot
- Line 74 ammoniacal system, line 75 ammonium system – please unify (decide to one of this form), in the rest of the article appears several more times
Forms to “ammoniacal system” were corrected throughout the text. Thanks for the gesture.
- Temperature once in K once in oC, sometimes in this both, there is suggestion to use only K, the same concerns concentration, mass and time in the whole article (using this two unit is chaotically)
The temperatures of ° C, concentrations and masses different from those used in the results of the own investigation are shown in the chapter of "Introduction". We cannot eliminate this way of showing the units, since they are original of the referenced articles, however, the units are converted to the units that we show here, in our work. Those conversions are in parentheses. Thanks anyway for your suggestions.
- Line 74 – chapter needs correction (style)
Style corrected. Thank you.
- Line 95 (stirring rate – rev/min or rpm
We have left it throughout the text as “rev / min”. Thank you.
- Line 103 – malachite solution? Or copper solution
The referee is right. It was changed to "copper solution". Thanks for the gesture.
- Line 116 – Fe2+ passed to Fe3+
That's right, it was oxidized to Fe3+. Corrected in the text. Thank you.
- Line 111 Arzutug [12]
The reference number was added in the text. Thank you.
- Line 120 – in this work [13]
The way of referencing was corrected. Thank you very much for this warning.
- Line 124 – taguchi method in Kursuncu [14]
Added in the text “… .in KurÅŸuncu et al. " Thank you.
- Line 126-127 – sentence should be rearranged
The sentence was arranged. Thanks for the suggestion.
- Line 137 – 138 – sentence not clear, should be rearranged
We believe that the sentence is correct. Thank you.
- Line 145 - sentence not clear, should be rearranged
The sentence was arranged. Thanks for the suggestion.
- 1 – concentration of cooper or coper ions
It was not understood what should be corrected.
- Line 176-180 – not clear – should be rearranged
The sentence was changed. It was better for understanding. Thanks for the gesture.
- Line 187-188 – not clear – should be rearranged
The sentence was modified. Thank you.
- Line 197 - not clear – should be rearranged
We believe that it is not necessary to add the reaction on carbonates and water, since we can reference it (and thus leave fewer reactions in the text). Thanks for understanding.
- Line 210 - not clear – should be rearranged
We believe that the explanation is very similar to that given for ammoniacal leaching. Thank you very much for understanding.
- Line 218 chapter 3.3. the same mark as in pkt 2 (ammoniacal or ammonium)
It was changed to "Ammoniacal leaching". Thank you.
- In chapter 3 – there should be scheme of research – it will improve the visibility of research methodology
We agree with the referee. We were also thinking of a research scheme. However, at the suggestion of other researchers, we are in the current way of showing the experimental procedure, due to the low number of instruments and stages used in our research. Thank you very much for your suggestions on improving our manuscript.
- In Fig. 3 time, second
In Figure 3, it is effectively in seconds. Thank you.
- Line 325 – Figure 1 or Figure 5?
It is effectively referred to Figure 1. Thank you.
- Line 389-391 - not clear – should be rearranged
The sentence is settled. Thank you.
- Figure 5 – Cu(NH3)42 – there is need for +
The compound was fixed. Thanks for watching.
- Line 430 – 432 - not clear – should be rearranged
We find that the sentence is clear. It was even extracted and improved from the work of Oudenne. Thanks for the comment.
- Figure 6 , 7, 8, 9, 11 - black boxes appear in the drawings, what they mean, if anything, they need to be removed
When the document is in .word format, you don't have those areas, however when you transform it to .pdf format these errors occur and generate. Must be own printing. How to proceed will be analyzed. Thank you.
- Line 543-544 – not clear – wrongly formulated
We have analyzed the sentence, and it is clear. We have also analyzed the numbers (extraction) and they are within what is shown in the graph. Thanks for your comments.
- Figure 10 - a better solution would be to present the results in the form of a block diagram
That diagram had been thought of as well as others. However, the author has already published similar results, and in the metallurgical field this way of showing those results is widely used (interesting to analyze the interaction of the volume of solution). Thank you very much for your details and comments on the manuscript.
- Line 705 – not clear
That sentence has already been reviewed. Thank you.
- Figure 13 - reference is made to Fig. 6 while in line 731, Figure 8
The referee is right. Changed the reference given for figure 13. Thank you very much for the referee's time spent reviewing our manuscript. Our sincere thanks.

Reviewer 3 Report
Please see attachment

Author Response
Dear Reviewer,
Thank you very much for helping us to generate and improve this research manuscript. It is very important for us to have valuable comments from an expert in the field. Below we send our changes and / or suggestions: (in blue color)
- The authors should explain why they chose pH 6.0, 10.5 and 13.0. What about pH 9.5 or 11.5?
We think that the graph given in Figure 5 would explain the choice of pH values well. Furthermore, it is also mentioned in the text that the same author has used similar systems and equal pH values, for reasons of keeping copper tetraamine in solution. Thanks for the observation
- page 13, Figure 10: In my opinion, the difference of copper recovery for 0.4 dm3/kg (70%) and 0.6 dm3/kg (72%) is insignificant (probably in the limits of error). Was statistical analysis performed?
The referee is right. Most likely, it is not very relevant between the two generated values. Statistical analysis was not performed. The text explains that over 0.6 dm3 / kg, similar values of copper recovery are maintained, the main objective of this point. Thank you.
- page 13, chapter 8. Effect of different ammonium reagents: studies with NH4Cl would be a good complement to the presented results. Did the authors study malachite dissolution in NH4Cl?
The referee is right. NH4Cl is a good compound to be analyzed. It contributes NH4 to maintain copper, and Cl to accelerate the dissolution of compounds. However, in this study we did not evaluate it. For future research, we will consider it. Thanks for your suggestions.
Minor issues:
- page 5, lines 204 and 209: the same chapter names,
The referee is right. The name of the chapter “3.2 Acid tests” was changed. Thanks for the observation
- Figures 3, 6, 7, 8, 9, 11, 12, 13, 15 (X-axis): is “Time, seconds” but it should be “Time, s”,
We prefer to keep time in seconds. Thank you.
- "rev/min" should be replaced with “rpm”,
We prefer to keep stirring at “rev /min”. Thank you.
- there should be a space between the text and the reference: is “… inviable[1-2].” but it should be “… inviable [1-2].”. Please, correct it in the paper.
All spaces between the word and the reference found in the text were corrected. Thank you very much for the referee's suggestions to improve our manuscript. Thank you very much for your time, your space on the agenda and your exclusive dedication to further improve our work.

Round 2
Reviewer 1 Report
The reviewer understands the authors' comments. He will not argue about them if the authors are intendedly claiming.
Well, what do the shadowed regions in Fig. 6, 7, 8, 9, and 11 mean? If they originate in the compatibility of software only on the reviewer's laptop and other people are not annoyed, the authors may ignore this comment.
Author Response
- Well, what do the shadowed regions in Fig. 6, 7, 8, 9, and 11 mean? If they originate in the compatibility of software only on the reviewer's laptop and other people are not annoyed, the authors may ignore this comment.
Dear referee, I don't know what happens to the format of the images. When putting those images in the original document (word format), they appear without shadows. Anyway, what we have done is cut and paste those images using a basic platform tool. We hope that you solve the problem. Thank you.
